# Selectivity of Insecticides to a Pupal Parasitoid, *Trichospilus diatraeae* (Hymenoptera: Eulophidae), of Soybean Caterpillars

**DOI:** 10.3390/insects14030217

**Published:** 2023-02-22

**Authors:** Helter Carlos Pereira, Fabricio Fagundes Pereira, Vitor Bortolanza Insabrald, Augusto Rodrigues, Jéssica Terilli Lucchetta, Farley William Souza Silva, Winnie Cezario Fernandes, Zenilda de Fatima Carneiro, Pedro Henrique Breda Périgo, José Cola Zanuncio

**Affiliations:** 1Departamento de Ciências Agrárias, Universidade Federal da Grande Dourados, Dourados 79804, Brazil; 2Departamento de Ciências Biológicas e Ambientais, Universidade Federal da Grande Dourados, Dourados 79804, Brazil; 3Centro de Ciências Biológicas e da Natureza, Universidade Federal do Acre, Rio Branco 69915, Brazil; 4Departamento de Ciências Agrárias, Universidade Tecnológica Federal do Paraná, Pato Branco 85503, Brazil; 5Departamento de Entomologia/BIOAGRO, Universidade Federal de Viçosa, Viçosa 36570, Brazil

**Keywords:** Eulophidae, IPM, mortality, parasitoid, pesticide, selectivity

## Abstract

**Simple Summary:**

Lepidoptera caterpillars are important pests around the world that decrease crop production, including that of soybeans. The use of insecticides and bioinsecticides is the main control strategy for these pests. Biological control with parasitoids, in combination with insecticides, is widely used as an alternative. However, the use of insecticides with parasitoids depends on tests to assess the survival/mortality of these natural enemies for each insecticide. *Trichospilus diatraeae* Cherian & Margabandhu, 1942 (Hymenoptera: Eulophidae) is a polyphagous pupal parasitoid that mainly comprises defoliating lepidopterans. This study evaluated the mortality of *T. diatraeae* from seven insecticides (acephate, azadirachtin, *Bacillus thuringiensis* (*Bt*), deltamethrin, lufenuron, teflubenzuron and thiamethoxam + lambda-cyhalothrin). The insecticides were sprayed on soybean leaves, which were left to dry and placed in cages with 10 *T. diatraeae* females. The insecticides azadirachtin, *Bt*, lufenuron and teflubenzuron did not affect *T. diatraeae* survival; deltamethrin and thiamethoxam + lambda-cyhalothrin presented low toxicity; and acephate was highly toxic, causing 100% mortality in this parasitoid. Azadirachtin, *Bt*, lufenuron and teflubenzuron are selective for *T. diatraeae*.

**Abstract:**

Selectivity is an important aspect of modern insecticides to be able to target pests whilst maintaining beneficial entomofauna in the crop. The present objective was to assess the selectivity of different insecticides for the pupal parasitoid of soybean caterpillars, i.e., *Trichospilus diatraeae* Cherian & Margabandhu, 1942 (Hymenoptera: Eulophidae). Acephate, azadirachtin, *Bacillus thuringiensis* (*Bt*), deltamethrin, lufenuron, teflubenzuron and thiamethoxam + lambda-cyhalothrin at the highest recommended concentrations for the soybean looper *Chrysodeixis includens* (Walker, [1858]) (Lepidoptera: Noctuidae), as well as water in the control, were used against the pupal parasitoid *T. diatraeae*. The insecticides and the control were sprayed on the soybean leaves, which were left to dry naturally and placed in cages with *T. diatraeae* females in each one. Survival data were submitted to analysis of variance (ANOVA) and the means were compared using Tukey’s HSD test (α = 0.05). Survival curves were plotted according to the Kaplan–Meier method, and the pairs of curves were compared using the log-rank test at 5% probability. The insecticides azadirachtin, *Bt*, lufenuron and teflubenzuron did not affect *T. diatraeae* survival, while deltamethrin and thiamethoxam + lambda-cyhalothrin presented low toxicity and acephate was highly toxic, causing 100% mortality in the parasitoid. Azadirachtin, *Bt*, lufenuron and teflubenzuron are selective for *T. diatraeae* and could be used in IPM programs.

## 1. Introduction

Soybean monoculture (*Glycine max* (L) Merrill) favors the reproduction and development of insect pests, particularly lepidopteran caterpillars that feed on leaves, flowers and fruits, and that reduce crop productivity [1]. The use of insecticides is the main method to manage defoliating insects in soybeans, with lower initial costs, easy application and high efficiency [2]. However, integrated pest management (IPM) recommends the combined use of different control methods [3], including chemicals [4]. Selective products that cause high pest mortality and low impact on beneficial insects are preferable [5].

The insecticide acephate with the mechanism of inhibiting the action of acetylcholinesterase enzymes increases the nervous impulses, causing hyperexcitation and insect death. Deltamethrin is an insecticide from the group of sodium channel modulators. Its action mechanism causes the uncontrolled release of intracellular calcium, hyperexcitation, paralysis and insect death. Lufenuron and teflubenzuron are insecticides from the group of chitin biosynthesis inhibitors, preventing the growth and development of insects with a mechanism of action that mainly causes failure or accelerating ecdysis. Thiamethoxam + lambda-cyhalothrin is a mixture with the first as a neonicotinoid insecticide from the group of competitive modulators of nicotinic acetylcholine receptors and an acetylcholine-imitating mechanism, which stimulates the nerve cells and causes nervous hyperexcitation and insect death. The second, a pyrethroid from the group of sodium channel modulators, has an action mechanism that blocks sodium channels [6,7]. Azadirachtin, an insecticide obtained from the *Azadirachta indica* A. Juss plant, from the group of compounds with an unknown or uncertain mode of action is used as an alternative to synthetic insecticides. *Bacillus thuringiensis*, a biological insecticide from the group of microbial disruptors of the midgut membrane, is formed by bacteria that release crystals that dissolve into toxic proteins in the intestine following ingestion, destroying the midgut membrane and causing generalized infection (septicemia) and insect death. Bioinsecticides, such as azadirachtin and *Bacillus thuringiensis* (*Bt*), that present lower ecological harm and toxicity to non-target organisms, as well as faster breakdown, can complement the use of synthetic insecticides [8,9].

The polyphagous parasitoid *Trichospilus diatraeae* Cherian & Margabandhu, 1942 (Hymenoptera: Eulophidae) is used in biological control programs in the soybean crop, parasitizing pupae of Lepidopteran pests [10], including *Anticarsia gemmatalis* (Hübner, 1818) (Lepidoptera: Erebidae) [11,12], *Helicoverpa armigera* (Hübner, 1805) [10], *Spodoptera frugiperda* (J.E. Smith, 1797) and *Heliothis virescens* (Fabricius, 1781) (Lepidoptera: Noctuidae) [13]. *Trichospilus diatraeae* was first obtained from pupae of *Diatraae venosata* (Walker 1863) and described in South India in 1942 [14].

The high parasitism rate and low specificity yield mass rearing of *T. diatraeae* in pupae of different hosts [15]. Furthermore, this parasitoid may naturally be present in the field as an important agent in applied biological control programs [12].

The effects of insecticides on *T. diatraeae* are poorly understood, highlighting the need for further research for their use in conservative biological actions and/or inundative control to increase the success of IPM [16].

The objective of the study was to evaluate commercial insecticides in terms of their impact on the pupal parasitoid *Trichospilus diatraeae* Cherian & Margabandhu, 1942 (Hymenoptera: Eulophidae), a natural enemy of soybean caterpillars. 

## 2. Materials and Methods

### 2.1. Trichospilus diatraeae

*Trichospilus diatraeae* were obtained from a colony maintained in the “Laboratório de Controle Biológico de Insetos (LECOBIOL)” at the Universidade Federal da Grande Dourados (UFGD), Dourados, Brasil (22°11′56.7″ S 54°56’′00.5″ W). Adult insects were kept in glass tubes (length 8.5 cm; diameter: 2.5 cm), covered with cotton, and fed with a droplet of pure honey. One *D. saccharalis* pupa (natural host of the parasitoid) was exposed for 24 h for parasitism by five *T. diatraeae* females (48–72 h age) at 25 ± 2 °C, a relative humidity (RH) of 70 ± 10%, and for 14 h photophase (light hours) in a climate-controlled chamber. The emerged parasitoids were fed with honey droplets and allowed to mate for 48–72 h. After mating, females were isolated and used in the experiments [15].

### 2.2. Assessment of Pesticides Selectivity under Extended Laboratory Conditions (ASPECLE)

The ASPECLE System was adapted by Sanomia 2020 according to the International Organization for Biological Control (IOBC) and is a standard test used in experiments of insecticide selectivity to parasitoids [17].

Glass cylinders (Borosilicate, Laborglas™, São Paulo, São Paulo state, Brazil) that were 3.5 cm in diameter and 25 cm long were sealed with voile fabric and a perforated plastic cap was placed at both ends. A tube was inserted at one end for the ventilation system. The ventilation system consisted of a central tube connected to a compressor/vacuum cleaner (maximum vacuum 695 mmhg) (Dia-Pump Fanem™ vacuum pump, São Paulo, SP, Brazil) with 24 tubes (12 on the left and 12 on the right) where the glass cylinders were coupled. The ventilation system and the pump facilitated gas exchange, reducing the accumulation of toxic gases in the cylinders (Figure 1) [18].

### 2.3. Trichospilus diatraeae Exposure to Insecticides

Five chemical insecticides and two bioinsecticides, recommended for the management of defoliating caterpillars in soybean crop, were used at the maximum concentrations recommended by the manufacturer (Table 1).

The insecticides/bioinsecticides were diluted in water at the maximum concentration indicated by the manufacturer and were sprayed onto soybean plants at the V4 phenological stage (three fully developed trefoils or four nodes), following the methodology of the IOBC [19], with a hand pressure sprayer (Brudden Practical 2000^®^, 1.5 L). Soybean seeds were sown in 7 L pots with 2/3 soil (distroferric red Latosol) and 1/3 organic matter (chicken manure) and were kept in a greenhouse until phenological stage V4.

The products were applied on the plants until leaf runoff. The plants were then left to dry in the shade in an open environment. Soybean plants were collected and separated per treatment and taken to the laboratory with five leaves per plant per cage. Ten 48 h old *T. diatraeae* females were placed inside each cage with a piece of cotton moistened with water.

*T. diatraeae* forages in the field in search of hosts, flying over plants and walking on leaves [15]. At this moment, *T. diatraeae* makes contact with leaves sprayed with insecticides.

### 2.4. Trichospilus diatraeae Mortality

Groups of 10 *T. diatraeae* females that were 48 h old, mated and fed were placed in 12 cylinders, totaling 120 females per treatment. The number of dead individuals was counted 24 h after the experiment was completed.

Insecticides were classified according to IOBC standards, based on the 24 h laboratory mortality data as follows: Class I: innocuous (<30%); Class II: slightly harmful (30–79%); Class III: moderately harmful (80–99%); and Class IV: harmful (>99%) [20].

The mortality data of the parasitoid *T. diatraeae* were submitted for analysis of variance (ANOVA) and the means were compared using Tukey’s HSD test (α = 0.05) with the statistical program SASM-AGRI [21].

### 2.5. Trichospilus diatraeae Survival

A second group of 10 *T. diatraeae* females that were 48 h old, mated and fed were placed in 12 cages, totaling 120 females per treatment. This experiment lasted 120 h and the numbers of live individuals were counted daily. Parasitoids were considered dead if they remained immobile when touched with a brush.

Survival data were submitted to the SAS Proc LIFETEST [22] to estimate their means, which were compared in pairs using the log-rank test (α = 0.05). Data were used in the *T. diatraeae* survival curves with the Kaplan–Meier method using the Sigma Plot 10.0 program. 

## 3. Results

When exposed to azadirachtin, *Bt*, lufenuron, teflubenzuron and control, *T. diatraeae* showed 0% mortality. Exposure to acephate, deltamethrin and thiamethoxam + lambda-cyhalothrin resulted in 100, 41 and 60% mortality, respectively (Table 2).

Survival of *T. diatraeae* following exposure to the control, *Bt*, azadirachtin, lufenuron, teflubenzuron, deltamethrin, thiamethoxam + lambda-cyhalothrin and acephate was 72, 66, 69, 77, 80, 20, 0 and 0%, respectively (Figure 2).

The survival period of the parasitoid *T. diatraeae* varied (*p* < 0.05) between treatments (χ² = 953.86; GL = 7; *p* < 0.0001) (Figure 2) at 0, 114.5, 112.6, 67.8, 115.4, 118.8, 36.8, and 115.2 h with the insecticides acephate, azadirachtin, *Bacillus thuringiensis*, deltamethrin, lufenuron, teflubenzuron, thiamethoxan + lambda-cyhalothrin, and the control, respectively, until the last evaluation at 120 h (Figure 3).

## 4. Discussion

Zero mortality of *T. diatraeae* female after 24 h classifies azadirachtin as non-toxic to this parasitoid [20]. This insecticide is a tetranortriterpenoid and, with its derivatives, is used to manage agricultural pests through repellence, antifeeding or interfering with the development of female sexual gonads, reducing or preventing oviposition [9,23]. In addition, the observed zero mortality of *T. diatraeae* may be related to contamination because azadirachtin mainly acts through ingestion [16] and the parasitoid only had contact with residues of this insecticide on soybean leaves. The similar zero mortality of *T. diatraeae* females when exposed to *Bacillus thuringiensis* kurstaki also classifies this insecticide as non-toxic [20]. This bacterium is specific to Lepidopteran caterpillars and, after ingestion, it releases its toxins, which bind to receptors in the gut wall, breaking it and causing septicemia [24]. *Trichospilus diatraeae* survival is due to the high specificity of *B. thuringiensis* var. kurstaki to Lepidopterans and because it is unlikely that this parasitoid will ingest dry residues of this bacteria sprayed on soybean leaves [8]. The zero mortality of *T. diatraeae* females after 24 h classifies the insecticides lufenuron and teflubenzuron, from the benzoylurea group, as non-toxic to this parasitoid [20]. These insecticides do not affect adult insects as they are growth regulators inhibiting chitin synthesis [25]. This is similar to that reported for the insecticides lufenuron and novaluron from the same IRAC-appointed insecticide action group, specific to insect larvae and nymphs with the parasitoid *Diachasmimorpha longicaudata* (Ashmead, 1905) (Hymenoptera: Braconidae) [16,26]. The 100% mortality of *T. diatraeae* females after 24 h exposure to acephate classifies this insecticide as toxic [20]. This insecticide, like other organophosphates, inhibits acetylcholinesterase (AChE) in the nervous system, causing paralysis and insect death [27], similar to the insecticide malathion for the parasitoid *Palmistichus elaeisis* Delvare & La Salle, 1993 (Hymenoptera: Eulophidae) [28]. *Trichospilus diatraeae* female mortality within 24 h from deltamethrin and thiamethoxam + lambda-cyhalothrin classifies these insecticides as mildly toxic to this natural enemy [20]. These insecticides, from the pyrethroid and neonicotinoid groups, act by keeping sodium channels open, causing hyperexcitation and blocking the insect’s nervous system and or binding to acetylcholine (ACh), transforming it into nAChRs and causing diverse symptoms, including hyperexcitation, lethargy and paralysis [26]. Pyrethroids and neonicotinoids act on the nervous system and respiration at all stages of insect development [27,29].

After 120 h, with exposure to *Bt*, azadirachtin, lufenuron and teflubenzuron, may be due to the action mode of these insecticides and the behavior of this parasitoid [30]. *Bt* is more specific to a group of insects (lepidopteran caterpillars), and lufenuron and teflubenzuron only act during the early stages of insect development [9,24,25]. After 120 h, *T. diatraeae* female mortality in these treatments may be related to the natural mortality of this parasitoid, with an average longevity of 8.28 ± 1.01 days [15]. The death of all parasitoids with the insecticide acephate within 24 h confirms the high toxicity of this insecticide [16]. In addition, the lower survival with insecticides deltamethrin and thiamethoxam + lambda-cyhalothrin is due to the persistence of their dry residues on the leaves and the direct contact of the parasitoid, increasing its mortality over time [26,31]. 

The greater *T. diatraeae* female survival after 24 h of exposure confirms the safety of these insecticides for this parasitoid [32]. Botanical insecticides such as azadirachtin are considered safe for natural enemies, not causing significant mortality of the parasitoids *Trichogramma nubilale* Ertle & Davis, 1975 and *Trichospilus pupivorus* Ferrière, 1930 (Hymenoptera: Eulophidae) [33,34]. Even when *Bt* was ingested by chewing the chorion of the egg on which the insecticide was sprayed during parasitoid emergence, this did not cause mortality or reduce the survival of the parasitoid *Trichogramma pretiosum* Riley, 1879 (Hymenoptera: Trichogrammatidae) [8]. Lufenuron and teflubenzuron, from the benzoylurea group, are growth regulators in insects and do not affect the survival of adult natural enemies [32]. However, acephate is an organophosphate from a group of insecticides, generally with a broad spectrum and low selectivity, causing high mortality of parasitoids [35]. The lethal effects of the residues of deltamethrin [36] cause reduced survival of the parasitoid T. diatraeae after 120 h. This is similar to that reported for *Trichopria anastrephae* Lima, 1940 (Hymenoptera: Diapriidae), with a mean survival of 10% after 120 h of exposure to this insecticide, and *Pachycrepoideus vindemmiae* (Rondani, 1875) (Hymenoptera: Pteromalidae) with 100% mortality [27]. However, deltamethrin did not reduce *P. elaeisis* survival [36]. The reduced *T. diatraeae* survival rate with thiamethoxam + lambda-cyhalothrin is due to its toxicity as reported for the mortality of all *Telenomus podisi* Ashmead, 1893 (Hymenoptera: Platygastridae) individuals due to direct contact with insecticide residues [37].

## 5. Conclusions

Insecticides *Bacillus thuringiensis*, azadirachtin, lufenu ron and teflubenzuron were considered harmless, deltamethrin and thiamethoxam + lambda-cyhalothrin were considered low toxicity and acephate was considered toxic to females of parasitoid *T. diatraeae*. The latter should not be used with this parasitoid in pest management programs. Azadirachtin, *Bt*, lufenuron and teflubenzuron are selective to conserve *T. diatraeae* populations.

Because no sublethal impacts were observed, further testing with these chemicals is recommended to evaluate their effect on the longevity, fertility, mating and foraging behavior of this parasitoid prior to inclusion in IPM programs.

## Figures and Tables

**Figure 1 insects-14-00217-f001:**
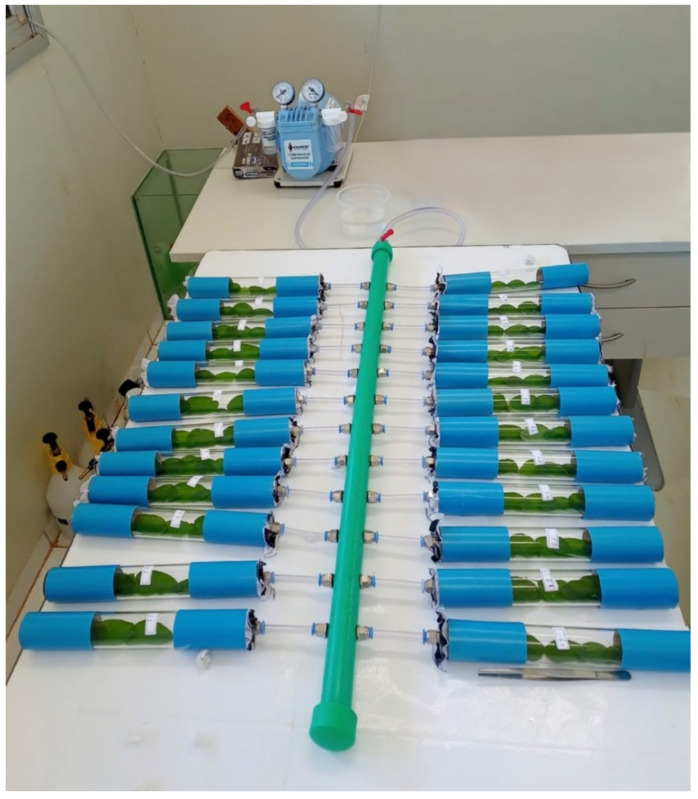
The ASPECLE System is a standard test used to assess insecticide selectivity to parasitoids.

**Figure 2 insects-14-00217-f002:**
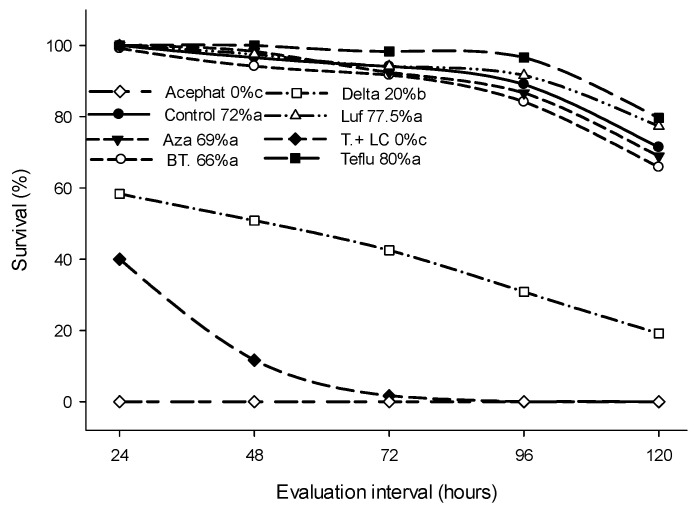
Mean percent survival (%) of *Trichospilus diatraeae* (Hymenoptera: Eulophidae) females after exposure to the insecticides acephate (acephat), azadirachtin (Aza), *Bacillus thuringiensis* (*Bt*), deltamethrin (Delta), lufenuron (Luf), teflubenzuron (Teflu), thiamethoxam + lambda-cyhalothrin (T. + LC) and to the control (Water). Pairs are compared survival curves estimated by the Kaplan–Meier method and the log-rank test (*p* < 0.05).

**Figure 3 insects-14-00217-f003:**
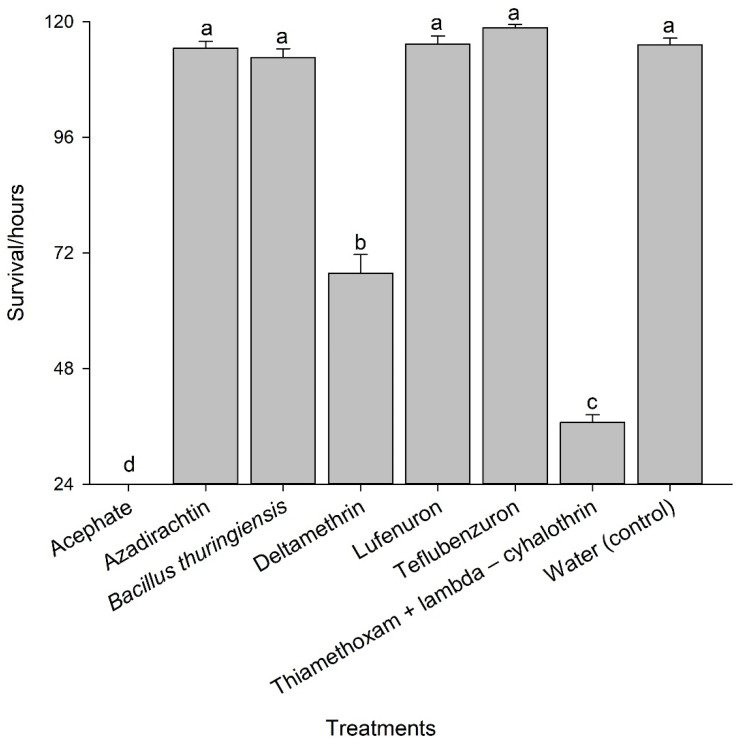
Survival, in hours (mean +/− SE), of *Trichospilus diatraeae* females after exposure to seven insecticides and the control. Bars with same letter are not significantly different (*p* < 0.05).

**Table 1 insects-14-00217-t001:** Active ingredients (AI), trade names (TN), chemical group, commercial product (CP), active ingredient concentration (AI) and toxicological classification (TC) of the insecticides acephate, azadirachtin, *Bacillus thuringiensis* (*Bt*), deltamethrin, lufenuron, teflubenzuron (Teflub.) and thiamethoxam + lambda-cyhalothrin (T. + Lc), recommended for the management of defoliating caterpillars in soybean [6].

AI	TN	Chemical Group	CP ^1^	AI ^2^	TC ^3^
Acephate	Acefato Nortox^®^	Organophosphate	667 g	450 g	IV
Azadirachtin	Neem Citromax^®^	Tetranortriterpenoid	2000 mL	2.4 mL	IV
B.t.	Dipel WP^®^	Microbiological	500 g	16 g	V
Deltamethrin	Decis 25 EC	Pyretroid	400 mL	16 mL	IV
Lufenuron	MATCH^®^ EC	Benzoylurea	374 mL	18.7 mL	V
Teflub.	Nomolt^®^ 150	Benzoylurea	200 mL	30 mL	IV
T. + Lc.	Engeo Pleno™ S	Neo *. + Pyrethoid	200 mL	28.2 + 21.2 mL	IV

^1^ CP: Commercial product in g or mL per 200 L ha^−1^. ^2^ AI: Active ingredient in g or mL per 200 L ha^−1^. ^3^ TC: Toxicological class (IV: slightly toxic; V: product unlikely to cause acute harm). * Neo: Neonicotinoid.

**Table 2 insects-14-00217-t002:** Percent mortality (mean ± standard error) of *Trichospilus diatraeae* females when exposed to seven insecticides for 24 h and toxicity class (TC).

Treatments	Mortality (%) *	TC+
Acephate	100.0 ± 0.0 a	4
Azadirachtin	0.0 ± 0.0 d	1
*Bacillus thuringiensis*	0.0 ± 0.0 d	1
Control	0.0 ± 0.0 d	1
Deltamethrin	41.7 ± 4.5 c	2
Lufenuron	0.0 ± 0.0 d	1
Teflubenzuron	0.0 ± 0.0 d	1
Thiamethoxam + Lambda-cyhalothrin	60.0 ± 4.5 b	2

* Means followed by the same letter within the column were not significantly different, Tukey’s test, *p* ≤ 0.05. TC+ = toxicity class where 1 = innocuous (<30%), 2 = slightly harmful (30–79%), 3 = moderately harmful (80–99%) and 4 = harmful (>99%) [20].

## Data Availability

All data sets presented in this study are included in the article and can be made available by the authors upon reasonable request.

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
