# Peer review of "Selectivity of Insecticides to a Pupal Parasitoid, Trichospilus diatraeae (Hymenoptera: Eulophidae), of Soybean Caterpillars"

_insects, 2023, doi:10.3390/insects14030217_

Round 1
Reviewer 1 Report
This study assessed the lethal effect of several chemical and biological pesticides on the pupae parasitoid Trichospilus diatraeae and screened to obtain insecticides with very low lethal effects on the parasitic wasp. Such results seem to be instructive for integrated pest management, but I have some concerns about the rigor of the experimental methods, scientific value and presentation. Therefore, the manuscript need major revision (control missing in some experiments).
1) The title of the article is “Insecticide selectivity to the pupae parasitoid”. However, I don't think the study reflects the so-called selectivity, the authors actually evaluated the safety of several pesticides on the parasitic wasp in a very brief way, as there were no studies on dose and sublethal effects. Also, the article does not contain any data about the effects of the tested pesticides on the target pests, which may limit the authors' findings.
2) I have listed the following articles for authors to refer to for normative research methods.
① Nicolas Desneux, Axel Decourtye, and Jean-Marie Delpuech / The Sublethal Effects of Pesticides on Beneficial Arthropods / https://www.annualreviews.org/doi/abs/10.1146/annurev.ento.52.110405.091440 / Annual Review of Entomology
② Moosa Saber • Zahra Abedi / Effects of methoxyfenozide and pyridalyl on the larval ectoparasitoid Habrobracon hebetor / DOI 10.1007/s10340-013-0528-4 / J Pest Sci
3) Mortality test: I suggest to strongly justify and detail any part of the data analysis. The authors do not report any information on how they checked the distribution of the data set. I would expect appropriate statistical tests to check for normality and homogeneity before applying the ANOVA, and in case of negative response a transformation of the data, or non-parametric tests, or generalized linear models would be required.
4) The manuscript is required to improve grammar and readability.
Author Response
Please find below our detailed comments to the points raised by the referees. We have begun these with three asterisks for clarity.
1) The title of the article is “Insecticide selectivity to the pupae parasitoid”. However, I don't think the study reflects the so-called selectivity, the authors actually evaluated the safety of several pesticides on the parasitic wasp in a very brief way, as there were no studies on dose and sublethal effects. Also, the article does not contain any data about the effects of the tested pesticides on the target pests, which may limit the authors' findings.
***Done
2) I have listed the following articles for authors to refer to for normative research methods.
① Nicolas Desneux, Axel Decourtye, and Jean-Marie Delpuech / The Sublethal Effects of Pesticides on Beneficial Arthropods / https://www.annualreviews.org/doi/abs/ 10.1146/annurev.ento.52.110405.091440 / Annual Review of Entomology
② Moosa Saber • Zahra Abedi / Effects of methoxyfenozide and pyridalyl on the larval ectoparasitoidHabrobraconhebetor / DOI 10.1007/s10340-013-0528-4 / J Pest Sci
***Done
3) Mortality test: I suggest to strongly justify and detail any part of the data analysis. The authors do not report any information on how they checked the distribution of the data set. I would expect appropriate statistical tests to check for normality and homogeneity before applying the ANOVA, and in case of negative response a transformation of the data, or non-parametric tests, or generalized linear models would be required.
***Done
4) The manuscript is required to improve grammar and readability.
***Done
Reviewer 2 Report
This is an interesting study conducted under the laboratory conditions with a view to determine selectivity of selected pesticides against a pupal parasitoid of soybean caterpillar. The study has investigated lethal effects of pesticides only. However, it has some application in the IPM. Further study on the sublethal effects of selected insecticides are needed before the final recommendations for the open field use.
The email of the authors needs to be official emails not the private ones.
The title of the paper is not clear. The paper needs to be formatted according to Insects author’s guidelines. For example, Simple summary is missing. Also, the reference quoting need to be using numbering system in the narrative for example [1], [2]…..
The following are the specific comments/suggestions:
Line 1-3: Change the title to “Insecticide selectivity to a pupal parasitoid, Trichospilus diatraeae (Hymenoptera: Eulophidae), of soybean caterpillars”
Line 21-22: Change to “Objective of the study was to assess the insecticide selectivity to a pupal parasitoid, Trichospilus diatraeae Cherian & Margabandhu, 1942 (Hymenoptera: Eulophidae) of soybean caterpillars”
Line 26: Use pupal parasitoid instead of puape parastioid
Line 27: Delete onto
Line 29: Use Tukey’s instead of Tukey
Line 35: Insert three additional keywords: Selectivity, Mortality, and IPM
Line 51: Use Lepidopteran
Line 66: Insert tube’s length and diameter
Line 70: Confirm 14 h photophase as light
Line 79: Use tubing instead of hose
Line 80: Insert air pressure of the pump
Line 90: Use Bt consistently; Use Teflubenzuron
Line 94: Delete I-extremely toxic, II-highly toxic……
Line 98: Explain why pesticide concentration was used instead of pesticide recommended dose by the manufacturer.
Line 103: Delete International Organization for Biological and Integrated Control
Line 116-121: Use Roman categories for the Classes
Line 138: Delete International Organization for Biological and Integrated Control
Table 2: Use Roman categories for the Classes
Line 152-152: Italicize P
Line 155: Use Bt.
Figure 3: Replot this figure as it is confusing. I suggest plot the survival (%) on Y-axis and time duration on the X-axis.
Line 165: Correct spelling for anti-feeding
Line 167: Azadirachtin is a chemical which shows antifeeding effects in insects by the mode of chemoreception (primary effect) and also food consumption by the insects is relatively reduced (secondary effect). Is this true?
Conclusions: Because no sub-lethal impacts were determined. This study would recommend additional testing of these chemicals with a view to know the effect of chemicals on the longevity, fecundity, mating, and foraging behavior of this parasitoid before their inclusion to IPM programs.
Line 244: Insert DAS.

Author Response
Please find below our detailed comments to the points raised by the referees. We have begun these with three asterisks for clarity.
This is an interesting study conducted under the laboratory conditions with a view to determine selectivity of selected pesticides against a pupal parasitoid of soybean caterpillar. The study has investigated lethal effects of pesticides only. However, it has some application in the IPM. Further study on the sublethal effects of selected insecticides are needed before the final recommendations for the open field use.
The email of the authors needs to be official emails not the private ones.
***These are the emails authors use more frequently
The title of the paper is not clear.
***Done
The paper needs to be formatted according to Insects author’s guidelines. For example, Simple summary is missing. Also, the reference quoting need to be using numbering system in the narrative for example [1], [2]…..
***Done
The following are the specific comments/suggestions:
Line 1-3: Change the title to “Insecticide selectivity to a pupal parasitoid, Trichospilus diatraeae (Hymenoptera: Eulophidae), of soybean caterpillars”
***Done
Line 21-22: Change to “Objective of the study was to assess the insecticide selectivity to a pupal parasitoid, Trichospilus diatraeae Cherian &Margabandhu, 1942 (Hymenoptera: Eulophidae) of soybean caterpillars”
***Done
Line 26: Use pupal parasitoid instead of puapeparastioid
***Done
Line 27: Delete onto
***Done
Line 29: Use Tukey’s instead of Tukey
***Done
Line 35: Insert three additional keywords: Selectivity, Mortality, and IPM
***Done
Line 51: Use Lepidopteran
***Done
Line 66: Insert tube’s length and diameter
***Done
Line 70: Confirm 14 h photophase as light
***Done
Line 79: Use tubing instead of hose
***Done
Line 80: Insert air pressure of the pump
***Done
Line 90: Use Bt consistently; Use Teflubenzuron
***Done
Line 94: Delete I-extremely toxic, II-highly toxic……
***Done
Line 98: Explain why pesticide concentration was used instead of pesticide recommended dose by the manufacturer.
***It was used the highest dose recommended by the manufacturer.
Line 103: Delete International Organization for Biological and Integrated Control
***Done
Line 116-121: Use Roman categories for the Classes
***Done
Line 138: Delete International Organization for Biological and Integrated Control
***Done
Table 2: Use Roman categories for the Classes
***Done
Line 152-152: Italicize P
***Done
Line 155: Use Bt.
***Done
Figure 3: Replot this figure as it is confusing. I suggest plot the survival (%) on Y-axis and time duration on the X-axis.
***Done. We have presented the figure as seen in other papers published in Insects.
Example:
10.3390/insects13050443
10.1007/s10340-022-01481-9
10.1093/jee/tov273
Line 165: Correct spelling for anti-feeding
***Done
Line 167: Azadirachtin is a chemical which shows antifeeding effects in insects by the mode of chemoreception (primary effect) and also food consumption by the insects is relatively reduced (secondary effect). Is this true?
***According to the literature it is true. In the experiment, the insecticide was not applied on the parasitoid’s food.
Conclusions: Because no sub-lethal impacts were determined. This study would recommend additional testing of these chemicals with a view to know the effect of chemicals on the longevity, fecundity, mating, and foraging behavior of this parasitoid before their inclusion to IPM programs.
***Done
Line 244: Insert DAS.
***Done
Reviewer 3 Report
This manuscript documents the response of a pupal parasitoid to commonly used chemical pesticides. While the methods used are standard to determine their efficacy, there are some serious issues with the set up of the study, the description of the experiments and presentation of the results. Below I have documented the more serious of these. The English used throughout the manuscript could also benefit from someone familiar with scientific writing in English, not merely an English speaker. I started to correct the grammar then noticed the issue was throughout the entire manuscript.
L20 – ‘whilst keeping beneficial entomofauna’ – incomplete sentence
L21 – should be ‘pupal parasitoid’ throughout
L38 – ‘considerably reduce the productivity of soybean’
L38-40 – run-on sentence, split into smaller ones
Introduction needs more connection between the studies done and the natural exposure to these products in the field. Would the parasitoids only encounter the product after it was dried on the leaves?
Sections 2.4 and 2.5 – not clear how these experiments were carried out – if the mortality and survival were part of the experiment described in Section 2.3, then these are just data collected as part of the experiment – they aren’t separate experiments.
L114 – indicates that the number of dead individuals was counted 24 h ‘after experiment’, yet Figure 2 is showing percentage survival from 24-120 h post exposure to the products, which seems connected to Section 2.5. It is not clear how these experiments are different from each other.
Table 2 – spacing in table needs to be fixed
Figure 3 – this should be ‘Mean (+/- SE) number of hours T. diatraea survived when exposed to products and evaluated at intervals up to 120 hours.’ Or something like this.
Author Response
Many thanks for your complementary review of our manuscript now entitled "Selectivity of insecticides to a pupal parasitoid, Trichospilus diatraeae (Hymenoptera: Eulophidae), of soybean caterpillars” for Insects.
There were only minor points, and we believe we have addressed them all. We hope that you find these amendments and explanations satisfactory and would of course be willing to make any further changes necessary.
Please find below our detailed comments to the points raised by the referees. We have begun these with three asterisks for clarity.
This manuscript documents the response of a pupal parasitoid to commonly used chemical pesticides. While the methods used are standard to determine their efficacy, there are some serious issues with the set up of the study, the description of the experiments and presentation of the results. Below I have documented the more serious of these. The English used throughout the manuscript could also benefit from someone familiar with scientific writing in English, not merely an English speaker. I started to correct the grammar then noticed the issue was throughout the entire manuscript.
L20 – ‘whilst keeping beneficial entomofauna’ – incomplete sentence
***Done
L21 – should be ‘pupal parasitoid’ throughout
***Done
L38 – ‘considerably reduce the productivity of soybean’
***Done
L38-40 – run-on sentence, split into smaller ones
***Done
Introduction needs more connection between the studies done and the natural exposure to these products in the field. Would the parasitoids only encounter the product after it was dried on the leaves?
***Done
L114 – indicates that the number of dead individuals was counted 24 h ‘after experiment’, yet Figure 2 is showing percentage survival from 24-120 h post exposure to the products, which seems connected to Section 2.5. It is not clear how these experiments are different from each other.
***Done
Table 2 – spacing in table needs to be fixed
***Done
Figure 3 – this should be ‘Mean (+/- SE) number of hours T. diatraea survived when exposed to products and evaluated at intervals up to 120 hours.’ Or something like this.
***Done
Round 2
Reviewer 2 Report
The revised manuscript has been improved.
Author Response
We would like to thank the referee for the valuable suggestions.
Reviewer 3 Report
The authors have improved the manuscript in some areas, but further attention is required prior to it being ready for publication. Specifically, the authors flip between 'parasitoid' and 'natural enemy'. Better to use 'parasitoid' throughout the manuscript as you are referring to a specific natural enemy in this manuscript. The discussion needs to be reorganized for clarity. Currently, it is presenting the treatments in alphabetical order which, given their results showing 2 groups (safe and un-safe for T. diatraeae), is difficult to follow their train of thought as the authors are flipping back and forth between the groups. Specific comments and suggestions to further improve the manuscript are presented below.
Introduction – this is improved but could benefit from reorganization.
L54 – ‘which reduce crop productivity’. The next section (L55-59) could start on L54, no need for a new paragraph
L60-62 – this sentence would make more sense just above the section which starts on L81 ‘The polyphagous parasitoid…’
L63-80 – this section is an improvement over the first version, but the connection between the synthetic products and their impact on a wide range of Families/Orders should be presented first, with the details about Azadirachtin and Bt and how they are ‘softer’ coming at the end of this section. Lines 60-62 would then finish off this section.
L86 – start last sentence with ‘Trichospilus diatraeae’ rather than ‘The natural enemy’
L88 – ‘High parasitism rate and low host specificity’ are not the factors that allow mass rearing. These factors make T. diatraeae attractive as a biological control agent.
L89 – if the species is present naturally then that further increases the attraction of it as you are augmenting natural populations.
L94 – suggest rewording slightly… ‘The objective of the study was to evaluate commercial insecticides for impact on the pupal parasitoid, Trichospilus diatraeae….’
L101 – suggest adding geo-coordinates for the Universidade Federal, or a civic address.
L103 – ‘for parasitism by five’
L106 – ‘allowed to mate for 48-72 h.’
L109 – if the ASPECLE is an ‘adaptation’ of the IOBC – who did this adaptation? If not the authors of this paper then please provide a reference. It’s not clear if the IOBC bioassay is the ‘standard’ or if the ASPECLE is the ‘standard’. Please clarify.
Figure 1 – suggest rewording ‘…. Assess insecticide impact on Trichospilus diatraeae females’ The impact of the insecticide is what allows you to determine the selectivity.
Table 1 – Suggest adding the units to the CP and AI columns directly. The reader is left guessing which are grams and which are mL. Not sure you need to specify the volume of water used per hectare? This volume is subject to change with the crop and the relevant concentration is amount of product/hectare, how it’s diluted is for field application purposes only.
L128 – ‘for the management of’
L129 – ‘in soybean.’
L137 – here is where you would specify the volume of water used per plant. If you were targeting a rate of 200L/ha, you should indicate the volume (approximate) of solution applied to each plant. The 7-L pots have an area or use the spray area of the hand sprayer as your ‘acreage’ and work backwards to the volume (at the 200L/ha rate) that should have been applied. At present it is difficult to know what concentration of product was applied to these potted plants. With the plants watered ‘until leaf runoff’ its difficult to know if more product than desired was applied.
L147 – suggest adding a sentence or two describing how T. diatraeae search for their hosts and what their exposure would be in the field. This will tighten the use of the treated leaves in the bioassay tubes. Yes, it is the ‘standard way’ to do these trials, but make the connection between the bioassay and the behavior of the parasitoid.
L149 – ‘Groups of 10 T. diatraeae females which were 48 h old, mated and fed…’
L150 – ‘counted 24 h after the experiment was completed’ – is this what you mean?
L159 – ‘A second group of 10 48 h hold T. diatraeae females which were mated and fed…’ Please clarify if the 10 groups in section 2.4 are the same as those used in 2.5 or if they are 2 distinct groups of parasitoids.
L160 – ‘This experiment lasted 120 h…’
L168 – needs rewording… ‘When exposed to azadirachtin, Bt, lufenuron, teflubenzuron and a control, T. diatraeae showed 0% mortality. Exposure to acephate, deltamethrin and thiamethozam + lambda-cyhalothrin resulted in 100, 41 and 60% mortality, respectively (Table 2).’
Table 2 – suggest rewording your caption ‘Percent mortality (mean ± SE) of Trichospilus diatraeae females when exposed to 7 insecticides for 24 h’ You don’t need the Order and Family in this caption, nor do you need IOBC here. Also suggest rounding to the nearest 10th, eg. 1 decimal place as the majority of your data is – and you’re dealing with a living organism. 100.0 +/- 0.0, 41.7 +/- 4.5. All the extra 0’s don’t add anything. What value is a ‘hundredth’ of a parasitoid? Footnote: need an ‘*’ after ‘Mortality (%)*’. ‘Means followed by the same letter within the column not significantly different, Tukeys’ test, P < 0.05. Put a ‘+’ by ‘TC+’, then in the footnote: +TC = toxicity class where 1 = innocuous, 2 = slightly harmful, 3 = …. Need to have the footnote and the table contents match (not 1 in table and I in footnote, or 2 in table and II in footnote)
L178 – reword please ‘Survival of T. diatraeae following 120 h exposure to a control, Bt, azadirachtin, lufenuron, teflubenzuron, deltamethrin, thiamethoxam+lambda-cyhalothrin and acephate was 72, 66, 69, 77.5, 80, 20, 0 and 0%, respectively (Figure 2).’ Aren’t these means? There were 12 groups of 10 females…. Shouldn’t these be means? Caption would then read ‘Mean percent survival of…’
Figure 2 – if the results at 120 h are the ones you are interested in, then why not simply present those as a bar graph? It would be easier to see the differences between the treatments that way. The graph presented here is better to show the progression over time. X-axis title ‘Hours of exposure’ would better represent the experiment. Suggest you move the significance letters to Figure 3.
L188 – need to ensure the reader knows these are means (see L178)
L189-190 – only 1 decimal place necessary for these values and no need to present the SE if these are captured in Figure 3.
Figure 3 – the experiment ended at 120 h but not all parasitoids survived to 120 h, so their exposure was less. Suggest rewording ‘Survival, in hours (mean +/- SE), of Trichospilus diatraeae females after exposure to 7 insecticides and a control. Bars with same letter not significantly different, P <0.05’. Remove ‘Order:Family’. Y-axis title: ‘Survival / hours’ is all you need here. X-axis need a title ‘Treatments’
L198 – ‘lack of mortality’ is not how best to say this…. ‘Zero mortality of T. diatraeae females after 24 h exposure to axadirachtin classifies this insecticide as non-toxic for this parasitoid…’
L200 – ‘or interfering with the development…’
L202 – ‘…the observed zero mortality of T. diatraeae may be related to….’
L204 – ‘Similar zero mortality of T. diatraeae females when exposed to Bacillus…’
L210 – ‘…unlikely that this parasitoid will ingest dry residues..’
L215 – ‘from the same group’ you mean the research group? Should specify this
L229 – delete ‘The’ start sentence with ‘After 120 h, high T. diatraeae female survival rates were observed with Bt, azadirachtin….and this may be due to the action mode…’
L231 – ‘specific to a group of insects’ – be more specific and clear with what you mean here
L238 – here you indicate ‘direct contact of the parasitoid’ which emphasizes the need to describe, even briefly, their host seeking behavior once on a plant so the reader can understand that the bioassay is indicative of the exposure they would experience in the field.
L240 – need to specify which treatments you are referring to here as this is a new section. And what is the connection with reference 31? Did they show the same result? If so, indicate that as it is difficult to know if you are referring to present results or only to what has already been observed.
L241 – not sure of the connection between L240 and 241 here. On L240 the section appears to be discussing the treatments that are safe for the parasitoid (use ‘parasitoid’ rather than ‘natural enemy’ throughout for consistency). Acephate, as described in L234-236 shows it to be unsafe for T. diatraeae, so this section L240-248 is confusing with the impact of Acephate presented early in the section.
I don’t think you need to address the treatments in alphabetical order in the discussion. Based upon your results they are now grouped into ‘safe’ and ‘unsafe’ and I’d recommend discussing them like this rather than alphabetically. If there is a means to connect the mode of action of the treatments with the behavior of the parasitoid, that would be good to include in the discussion to further explain the differences observed.
L246 – ‘even when ingested’? earlier in the MS you indicate that T. diatraeae did not likely ingest Bt (L209) – why would T. pretiosum ingest and T. diatraeae not? Can you provide further detail on how authors in ref [6] confirmed ingestion?
L250 – reference [35] are these your observations or observations from [35]? Try rewording ‘The reduced survival of T. diatraeae following exposure to deltamethrin confirms the lethal effects of these residues, a finding consistent with other studies [35].’
L265 – ‘selective’ could also indicate lethality… better to state ‘are selective to conserve T. diatraeae populations.’ Then it’s specific what the selectivity is for.
There were no references included with this version of the manuscript?
Author Response
L54 – ‘which reduce crop productivity’. The next section (L55-59) could start on L54, no need for a new paragraph
***Done
L60-62 – this sentence would make more sense just above the section which starts on L81 ‘The polyphagous parasitoid…’
***Done
L63-80 – this section is an improvement over the first version, but the connection between the synthetic products and their impact on a wide range of Families/Orders should be presented first, with the details about Azadirachtin and Bt and how they are ‘softer’ coming at the end of this section. Lines 60-62 would then finish off this section.
***Done
L86 – start last sentence with ‘Trichospilus diatraeae’ rather than ‘The natural enemy’
***Done
L88 – ‘High parasitism rate and low host specificity’ are not the factors that allow mass rearing. These factors make T. diatraeae attractive as a biological control agent.
***Done
L89 – if the species is present naturally then that further increases the attraction of it as you are augmenting natural populations.
***Ok
L94 – suggest rewording slightly… ‘The objective of the study was to evaluate commercial insecticides for impact on the pupal parasitoid, Trichospilus diatraeae….’
***Done
L101 – suggest adding geo-coordinates for the Universidade Federal, or a civic address.
***Done
L103 – ‘for parasitism by five’
***Done
L106 – ‘allowed to mate for 48-72 h.’
***Done
L109 – if the ASPECLE is an ‘adaptation’ of the IOBC – who did this adaptation? If not the authors of this paper then please provide a reference. It’s not clear if the IOBC bioassay is the ‘standard’ or if the ASPECLE is the ‘standard’. Please clarify.
***Done
Figure 1 – suggest rewording ‘…. Assess insecticide impact on Trichospilus diatraeae females’ The impact of the insecticide is what allows you to determine the selectivity.
***This figure is to demonstrate how is the device that was used for the experiment
Table 1 – Suggest adding the units to the CP and AI columns directly. The reader is left guessing which are grams and which are mL. Not sure you need to specify the volume of water used per hectare? This volume is subject to change with the crop and the relevant concentration is amount of product/hectare, how it’s diluted is for field application purposes only.
***Done. This measure 200L/ha (water + product) was used to standardize the recommendation that the Federal Government establishes in Brazil. We use as the maximum insecticide dosage that can be used in 200 liters of water in one hectare.
L128 – ‘for the management of’
***Done
L129 – ‘in soybean.’
***Done
L137 – here is where you would specify the volume of water used per plant. If you were targeting a rate of 200L/ha, you should indicate the volume (approximate) of solution applied to each plant. The 7-L pots have an area or use the spray area of the hand sprayer as your ‘acreage’ and work backwards to the volume (at the 200L/ha rate) that should have been applied. At present it is difficult to know what concentration of product was applied to these potted plants. With the plants watered ‘until leaf runoff’ its difficult to know if more product than desired was applied.
***200 L/ha A maximum dosage of spray (insecticide + water) that will be necessary to apply in 1 hectare of soybeans. Dosage may be lower depending on several factors. The purpose of the work is to simulate that a soybean area of 1 hectare requires a maximum amount of product. And therefore, the maximum dose of insecticide that could be applied was calculated according to the Brazilian Ministry of Agriculture. If the parasitoid can withstand the most adverse amount in the laboratory, theoretically it will be able to withstand a smaller amount in the field.
L147 – suggest adding a sentence or two describing how T. diatraeae search for their hosts and what their exposure would be in the field. This will tighten the use of the treated leaves in the bioassay tubes. Yes, it is the ‘standard way’ to do these trials, but make the connection between the bioassay and the behavior of the parasitoid.
***Done
L149 – ‘Groups of 10 T. diatraeae females which were 48 h old, mated and fed…’
***Done
L150 – ‘counted 24 h after the experiment was completed’ – is this what you mean?
***Done
L159 – ‘A second group of 10 48 h hold T. diatraeae females which were mated and fed…’ Please clarify if the 10 groups in section 2.4 are the same as those used in 2.5 or if they are 2 distinct groups of parasitoids.
***Done. Are 2 distinct groups of parasitoids.
L160 – ‘This experiment lasted 120 h…’
***Done
L168 – needs rewording… ‘When exposed to azadirachtin, Bt, lufenuron, teflubenzuron and a control, T. diatraeae showed 0% mortality. Exposure to acephate, deltamethrin and thiamethozam + lambda-cyhalothrin resulted in 100, 41 and 60% mortality, respectively (Table 2).’
***Done
Table 2 – suggest rewording your caption ‘Percent mortality (mean ± SE) of Trichospilus diatraeae females when exposed to 7 insecticides for 24 h’ You don’t need the Order and Family in this caption, nor do you need IOBC here. Also suggest rounding to the nearest 10th, eg. 1 decimal place as the majority of your data is – and you’re dealing with a living organism. 100.0 +/- 0.0, 41.7 +/- 4.5. All the extra 0’s don’t add anything. What value is a ‘hundredth’ of a parasitoid? Footnote: need an ‘*’ after ‘Mortality (%)*’. ‘Means followed by the same letter within the column not significantly different, Tukeys’ test, P < 0.05. Put a ‘+’ by ‘TC+’, then in the footnote: +TC = toxicity class where 1 = innocuous, 2 = slightly harmful, 3 = …. Need to have the footnote and the table contents match (not 1 in table and I in footnote, or 2 in table and II in footnote)
***Done
L178 – reword please ‘Survival of T. diatraeae following 120 h exposure to a control, Bt, azadirachtin, lufenuron, teflubenzuron, deltamethrin, thiamethoxam+lambda-cyhalothrin and acephate was 72, 66, 69, 77.5, 80, 20, 0 and 0%, respectively (Figure 2).’ Aren’t these means? There were 12 groups of 10 females…. Shouldn’t these be means? Caption would then read ‘Mean percent survival of…’
***Done
Figure 2 – if the results at 120 h are the ones you are interested in, then why not simply present those as a bar graph? It would be easier to see the differences between the treatments that way. The graph presented here is better to show the progression over time. X-axis title ‘Hours of exposure’ would better represent the experiment. Suggest you move the significance letters to Figure 3.
***Done. Some articles present the figure in this way.
Example: 10.1007/s10340-022-01481-9/ 10.1093/jee/tov273
The interesting thing about keeping the graph that way instead of a bar graph is to show that some of the non-selective insecticides mortality occurred over the course of days.
L188 – need to ensure the reader knows these are means (see L178)
***Done
L189-190 – only 1 decimal place necessary for these values and no need to present the SE if these are captured in Figure 3.
***Done
Figure 3 – the experiment ended at 120 h but not all parasitoids survived to 120 h, so their exposure was less. Suggest rewording ‘Survival, in hours (mean +/- SE), of Trichospilus diatraeae females after exposure to 7 insecticides and a control. Bars with same letter not significantly different, P <0.05’. Remove ‘Order:Family’. Y-axis title: ‘Survival / hours’ is all you need here. X-axis need a title ‘Treatments’
***Done
L198 – ‘lack of mortality’ is not how best to say this…. ‘Zero mortality of T. diatraeae females after 24 h exposure to axadirachtin classifies this insecticide as non-toxic for this parasitoid…’
***Done
L200 – ‘or interfering with the development…’
***Done
L202 – ‘…the observed zero mortality of T. diatraeae may be related to….’
***Done
L204 – ‘Similar zero mortality of T. diatraeae females when exposed to Bacillus…’
***Done
L210 – ‘…unlikely that this parasitoid will ingest dry residues..’
***Done
L215 – ‘from the same group’ you mean the research group? Should specify this
***Done
L229 – delete ‘The’ start sentence with ‘After 120 h, high T. diatraeae female survival rates were observed with Bt, azadirachtin….and this may be due to the action mode…’
***Done
L231 – ‘specific to a group of insects’ – be more specific and clear with what you mean here
***Done
L238 – here you indicate ‘direct contact of the parasitoid’ which emphasizes the need to describe, even briefly, their host seeking behavior once on a plant so the reader can understand that the bioassay is indicative of the exposure they would experience in the field.
***It was described in the last paragraph of item 2.3
L240 – need to specify which treatments you are referring to here as this is a new section. And what is the connection with reference 31? Did they show the same result? If so, indicate that as it is difficult to know if you are referring to present results or only to what has already been observed.
***Done
L241 – not sure of the connection between L240 and 241 here. On L240 the section appears to be discussing the treatments that are safe for the parasitoid (use ‘parasitoid’ rather than ‘natural enemy’ throughout for consistency). Acephate, as described in L234-236 shows it to be unsafe for T. diatraeae, so this section L240-248 is confusing with the impact of Acephate presented early in the section.
I don’t think you need to address the treatments in alphabetical order in the discussion. Based upon your results they are now grouped into ‘safe’ and ‘unsafe’ and I’d recommend discussing them like this rather than alphabetically. If there is a means to connect the mode of action of the treatments with the behavior of the parasitoid, that would be good to include in the discussion to further explain the differences observed.
***Done
L246 – ‘even when ingested’? earlier in the MS you indicate that T. diatraeae did not likely ingest Bt (L209) – why would T. pretiosum ingest and T. diatraeae not? Can you provide further detail on how authors in ref [6] confirmed ingestion?
***Done
L250 – reference [35] are these your observations or observations from [35]? Try rewording ‘The reduced survival of T. diatraeae following exposure to deltamethrin confirms the lethal effects of these residues, a finding consistent with other studies [35].’
***Done
L265 – ‘selective’ could also indicate lethality… better to state ‘are selective to conserve T. diatraeae populations.’ Then it’s specific what the selectivity is for.
***Done